



# Separating emission and meteorological contribution to PM$_{2.5}$ trends over East China during 2000–2018

Qingyang Xiao[1,#], Yixuan Zheng[2,#], Guannan Geng[1,3], Cuihong Chen[4,5], Xiaomeng Huang[4], Huizheng Che[6], Xiaoye Zhang[6], Kebin He[1,3], Qiang Zhang[4]

[1]State Key Joint Laboratory of Environment Simulation and Pollution Control, School of Environment, Tsinghua University, Beijing 100084, China
[2]Atmospheric Environment Institute, Chinese Academy of Environmental Planning, Beijing 100012, China
[3]State Environmental Protection Key Laboratory of Sources and Control of Air Pollution Complex, Beijing 100084, China
[4]Ministry of Education Key Laboratory for Earth System Modelling, Department of Earth System Science, Tsinghua University, Beijing 100084, China
[5]Satellite Environment Center, Ministry of Ecology and Environment of the People's Republic of China, Beijing 100094, China
[6]State Key Laboratory of Severe Weather & Key Laboratory of Atmospheric Chemistry of CMA, Chinese Academy of Meteorological Sciences, Beijing 100081, China

[#]These authors contributed equally to this work.

*Correspondence to*: Guannan Geng (guannangeng@tsinghua.edu.cn)

**Abstract.** The contribution of meteorology and emissions to long-term PM$_{2.5}$ trends is critical for air quality management but has not yet been fully analyzed. Here, we used a combination of machine learning model, statistical model and chemical transport model to quantify the meteorological impacts on PM$_{2.5}$ pollution during 2000–2018. Specifically, we first developed a two-stage machine learning PM$_{2.5}$ prediction model with a synthetic minority oversampling technique to improve the satellite-based PM$_{2.5}$ estimates over highly polluted days, thus allowing us to better characterize the meteorological effects on haze events. Then we used two methods, a generalized additive model (GAM) driven by the satellite-based full-coverage daily PM$_{2.5}$ retrievals as well as the Weather Research and Forecasting/Community Multiscale Air Quality (WRF/CMAQ) modelling system, to examine the meteorological contribution to PM$_{2.5}$. We found good agreements between the GAM model estimations and the CMAQ model estimations of meteorological contribution to PM$_{2.5}$ on monthly scale (correlation coefficient between 0.53–0.72). Both methods revealed the dominant role of emission changes in the long-term trend of PM$_{2.5}$ concentration in China during 2000–2018, with notable influence from the meteorological condition. The interannual trends in meteorology-associate PM$_{2.5}$ were dominated by the fall and winter meteorological conditions, when regional stagnant and stable conditions were more likely to happen and haze events frequently occurred. From 2000 to 2018, the meteorological contribution became more unfavorable to PM$_{2.5}$ pollution across the North China Plain and central China, but were more beneficial to pollution control across the southern part, e.g., the Yangtze River Delta. The meteorology-adjusted PM$_{2.5}$ over East China peaked at 2006 and 2011, mainly driven by the emission peaks in primary PM$_{2.5}$ and gas precursors in these years. Although emissions dominated the long-term PM$_{2.5}$ trends, the meteorology-driven anomalies also contributed –3.9% to 2.8% of the annual mean PM$_{2.5}$ concentrations in East China estimated from the GAM model. The meteorological contributions were even higher



regionally, e.g., –6.3% to 4.9% of the annual mean PM$_{2.5}$ concentrations in the Beijing-Tianjin-Hebei region, –5.1% to 4.3% in the Fen-wei Plain, –4.8% to 4.3% in the Yangtze River Delta and –25.6% to 12.3% in the Pearl River Delta. Considering the remarkable meteorological effects on PM$_{2.5}$ and the worsening meteorological conditions in the northern part of China where air pollution was severe and population was clustered, stricter clean air actions are needed to avoid haze events in the future.

## 1 Introduction

The air pollution, especially PM$_{2.5}$ pollution, has become a serious problem in China in the past decades. Variations in air pollution are primarily driven by two factors: emissions and meteorology. Anthropogenic emissions dominate the long-term trend of air pollution (Zhang et al., 2019a;Cheng et al., 2019a), and meteorological conditions also notably influence the daily, seasonal, interannual and interdecadal air pollution variations (Zhang et al., 2018;Chen et al., 2020;Wang et al., 2019a;Zhai et al., 2019). In China, changes in major air pollutant emissions attributable to the economic development and clean air policies have been widely studied (Guan et al., 2014;Shen et al., 2017). For example, during the 11$^{th}$ Five-Year Plan (2006–2010) and the 12$^{th}$ Five-Year Plan (2011–2015), gas pollutant emissions, i.e., SO$_2$ and NO$_x$, have been remarkably reduced (Ma et al., 2019;Geng et al., 2019). During "the Air Pollution Prevention and Control Action Plan" (The Action Plan, 2013-2017) and the Blue Sky Protection Campaign (2018–2020), PM$_{2.5}$ emissions dropped significantly and the PM$_{2.5}$ concentrations substantially decreased (Bian et al., 2019;Liu et al., 2015). Meanwhile, air pollution was also affected by the long-term trend of meteorological systems and climate change, especially in the context of global warming (Ruijin et al., 2017;Wang and Chen, 2016;Yi et al., 2019;Zhang et al., 2019b). Distinguishing the contributions of emission and meteorology is critical for the evaluation of clean air policies, projection of the future air quality and understanding pollution process.

Various methods have been reported to separate the contributions of emissions and meteorology. For example, chemical transport models (CTMs) simulate the atmospheric process with emission inventory and meteorology fields as inputs, thus allowing researchers to assess the changes in air pollution attributable to one factor when controlling another factor (Wang et al., 2019a;Xu et al., 2020;Zheng et al., 2017). CTM simulations have been widely used to separate the contributions of meteorology and anthropogenic emissions to air pollution variations. With appropriate study design, the CTM modelling system can reasonably assess the influence of a specific emission reduction measure or a specific meteorological condition on air pollution. However, these model simulations require considerable computation resources, and the quality of inputs (e.g., emission inventory and meteorology) affects the quality of simulations. Moreover, due to the interactions between emissions and meteorology, the simulations in the fixed emission scenarios and the fixed meteorology scenarios may not fully reflect real-world conditions.

Other studies have applied statistical methods to assess the meteorology-associate changes in air pollution and to quantify the contribution of emissions. Multiple linear regression (MLR) has been adopted to describe the relationships between





meteorology and air pollutant concentrations (Cheng et al., 2019a;Sá et al., 2015). For example, Zhai et al. (2019) constructed deseasonalized and deseasonalized-detrended time-series data and assessed the meteorological effects by MLR. Some studies also employed machine learning algorithms to better describe the non-linear relationships between meteorology and air pollution (Grange et al., 2018;Vu et al., 2019;Zhang et al., 2020). However, as such methods requires

continuous $PM_{2.5}$ data as inputs, previous studies relied on $PM_{2.5}$ ground measurements that were limited to certain locations (e.g., ground monitoring stations) and times (e.g., after 2013 in China). The limited sample size not only affected the model quality and introducing sampling bias, but also hampered the analyses on spatial heterogeneity of meteorology contributions across China. The relatively short study period failed to show the long-term trend of meteorology-associate $PM_{2.5}$. The analysis on the complete-coverage long-term trends of meteorology and emission contributions to air pollution is urgently

needed to support further evaluation of clean air policies and region-specific air quality management within the context of climate change.

In this study, we aimed to analyze the spatiotemporal trends in meteorology- and emission- associate $PM_{2.5}$ variations across China during 2000–2018. The meteorological impacts on $PM_{2.5}$ trends were assessed with data-fusion $PM_{2.5}$ predictions and chemical transport model simulations, taking advantage of the complete spatiotemporal coverage and long data records of

these two datasets. The data-fusion $PM_{2.5}$ predictions were derived by combining satellite data, chemical transport model simulations, ground measurements and ancillary data with an optimized two-stage machine model that improved the $PM_{2.5}$ estimates during highly polluted days. Then we assessed the long-term variations in meteorology-associate $PM_{2.5}$ using a generalized additive model (GAM) that better described the non-linear associations between $PM_{2.5}$ and meteorology. We also estimated the meteorological impacts on $PM_{2.5}$ trends with chemical transport model simulations under different scenarios

coupled with a most recent emission inventory. We showed that the temporal trends of meteorology-associate $PM_{2.5}$ estimated from the GAM method and from the chemical transport model were highly consistent. The trend analysis of the meteorology and emission contributions to $PM_{2.5}$ could support making of air quality management plans in the future.

## 2. Data and methods

This study employed simulations from the Weather Research and Forecasting/Community Multiscale Air Quality

(WRF/CMAQ) modelling system as well as gridded $PM_{2.5}$ predictions fused from multiple data sources to assess the meteorological effects on $PM_{2.5}$ (Fig. 1). The study domain covers East China (east of longitude 105°) and the $PM_{2.5}$ concentrations during 2000–2018 were analyzed.

### 2.1 Satellite-based $PM_{2.5}$ retrievals

Previously reported satellite-based $PM_{2.5}$ data tended to underestimate high pollution events (Xiao et al., 2018;Xue et al., 2019)

because these events rarely occurred in the model training dataset and were less characterized by the model. Since high pollution events were largely affected by meteorological conditions (Zhang et al., 2015;Liu et al., 2017b), correctly capturing



these events was critical for the assessment of meteorological contributions. Thus, we developed a two-stage model to improve the prediction accuracy of $PM_{2.5}$ estimates, especially over highly polluted days, and obtained spatiotemporally continuous daily $PM_{2.5}$ dataset during 2000–2018.

### 100   2.1.1 Data for $PM_{2.5}$ modeling

We assimilated the daily $PM_{2.5}$ measurements, WRF/CMAQ simulations, satellite aerosol optical depth (AOD) from Aqua and Terra MODIS Level 2 products (https://ladsweb.modaps.eosdis.nasa.gov/), meteorological parameters from the Modern-Era Retrospective analysis for Research and Applications Version 2 (MERRA-2) (Randles et al., 2017;Buchard et al., 2017), elevation data from the Global Digital Elevation Model (GDEM) (https://earthexplorer.usgs.gov/), gridded population

distributions (Xiao et al., 2021b), and land cover classification data (http://data.ess.tsinghua.edu.cn)(Gong et al., 2019a;Gong et al., 2019b) to train the $PM_{2.5}$ prediction model and predicted $PM_{2.5}$ concentrations during 2000–2018. The detailed data collection and processing methods were summarized in Appendix A.

### 2.1.2 The two-stage prediction model

A two-stage prediction model was developed to estimate $PM_{2.5}$ concentrations over China (Fig. 1). The first-stage model

described high-pollution events that were underestimated in previous models, in which a synthetic minority oversampling technique (SMOTE) was adopted (Torgo, 2010). The second-stage model predicted $PM_{2.5}$ concentrations with the high-pollution indicator from the first-stage model.

Since high-pollution events relatively rarely occur in the model training dataset, the model may not appropriately characterize the associations between high $PM_{2.5}$ concentrations and the predictors, leading to underestimation of high-

pollution levels (Wei et al., 2020). To balance high-pollution samples and normal samples, we first defined a high-pollution indicator, describing whether the daily $PM_{2.5}$ observation was higher than the monthly average $PM_{2.5}$ concentration plus two standard deviations at each location. A total of 3.9% of the daily data were assigned as high-pollution. Previous studies reported that balancing training data with SMOTE improved the classifiers' performance (Ghorbani and Ghousi, 2020;Saputra and Suharjito, 2019). Thus, we applied the SMOTE algorithm that oversampled the high-pollution data (the

minority) by artificially generated new synthetic samples along the line between the high-pollution data and their selected nearest neighbors (Chawla et al., 2002;Chawla et al., 2003). This method also under-sampled the normal data (the majority) to better balance the uneven proportion of the high-pollution and normal data. With SMOTE resampling, high-pollution data accounted for 23% in the new model training dataset.

The balanced model training dataset was adopted to train the first-stage extreme gradient boosting (XGBoost) model with all

the predictors, excluding CMAQ simulations. The predicted high-pollution indicator from the first-stage model was passed to the second-stage model as a predictor. We adopted the residual between the $PM_{2.5}$ measurement and the CMAQ $PM_{2.5}$





simulation as the dependent variable to train the second-stage model. Training the model with residual enhances the response of predictors to PM$_{2.5}$ variations, thus improved the prediction accuracy.

To fill any missing satellite data, in both the first- and second-stage model, we assigned the availability of satellite retrievals
as a dichotomous predictor and constructed it as the cutoff point of the first layer of the decision tree to separate the training data, thus mining the association between the availability of satellite retrievals and the PM$_{2.5}$ concentration. This method that fills missing PM$_{2.5}$ predictions with a decision tree outperformed other gap-filling methods in our previous evaluation study (Xiao et al., 2021a). The inclusion of CMAQ simulations also improved the accuracy of the gap-filled results.

The model's hyper parameter optimization and performance evaluation were conducted through five-fold CV, by-year CV
and by-location CV (Appendix B).

### 2.2 Assessment of the meteorological effects on PM$_{2.5}$ using GAM

Following the method described by Zhai et al. (2019), we constructed time-series data to distinguish the long-term, seasonal, and short-term trends of PM$_{2.5}$ concentrations and meteorological conditions. Then the associations between PM$_{2.5}$ and meteorology were fitted with a GAM model, using daily satellite-based PM$_{2.5}$ predictions as dependent variable. GAM has
been previously used to predict PM$_{2.5}$ concentrations with meteorology and other predictors (Yanosky et al., 2014;Liu et al., 2009;Xiao et al., 2018). The meteorological predictors in the GAM included 10-meter wind speed, 2-meter specific humidity, 2-meter air temperature, total precipitation, 10-meter eastward wind (U wind), 10-meter northward wind (V wind), U wind at 500 hpa, and planetary boundary layer height, which have been reported to be strongly associated with PM$_{2.5}$ concentrations in various regions in China (Chen et al., 2020).

Both the PM$_{2.5}$ data and the meteorology data followed the same processing protocol. First, we calculated 10-day average data, 50-day average data, and 19-year (2000–2018) average data based on the 50-day average data. We constructed deseasonalized-detrended data by removing the 50-day average data from the 10-day average data. We also constructed deseasonalized data by removing the 19-year average data from the 10-day average data. Assuming that the associations between PM$_{2.5}$ and meteorological parameters remained constant, we estimated these associations by a grid-specific seasonal
and year-round GAM model (Pearce et al., 2011) with the deseasonalized-detrended data. The GAM allows a nonlinear response of PM$_{2.5}$ levels to meteorological conditions, thus providing better fits to the training data (Table B1). We also fitted grid-specific seasonal stepwise MLR in a sensitivity analysis to examine whether the selection of model affects the assessment of meteorological effects. Additionally, normalized meteorological parameters were used to fit the linear regression. Hence, the estimated coefficients reflected the relative contribution of each meteorological parameter and
supported the spatial analysis of the meteorological effects. Since the seasonal model attained a higher average model R$^2$ than did the year-round model (Table B1), the results obtained with the seasonal model are presented in this study.





## 2.3 Assessment of the meteorological effects on PM$_{2.5}$ using WRF/CMAQ

We also used the WRF/CMAQ model to separate the contribution of emissions and meteorology on PM$_{2.5}$ trends. The CMAQ model version 5.1 driven by the WRF model version v3.5.1 were utilized in this study, and the model configurations were
160 following previous studies (Zheng et al., 2015, Zheng et al., 2017a). The initial and boundary conditions for WRF were derived from the National Centers for Environmental Prediction Final Analysis (NCEP-FNL) reanalysis data (NCEP, 2000). The boundary conditions for CMAQ were taken from the global GEOS-Chem model simulations. We used CB05 as the gas-phase mechanism, AERO6 as the aerosol module, and Regional Acid Deposition Model (RADM) as the aqueous-phase chemistry model in CMAQ. The anthropogenic emissions for mainland China were taken from the Multi-resolution Emission Inventory
of China (MEIC(Maji et al., 2019)(Maji et al., 2019)(Maji et al., 2019)(Maji et al., 2019), http://meicmodel.org/)(Zheng et al., 2018;Li et al., 2017a), and emissions beyond mainland China were from the MIX Asian emission inventory (Li et al., 2017b).

Two scenarios were conducted to estimate the meteorological impacts on PM$_{2.5}$ trends, the *BASE* scenario and the *FixEmis* scenario. The *BASE* scenario was simulated with year-by-year emissions and meteorology during 2000–2018, while the *FixEmis* scenario was conducted using fixed emissions at the 2000 level and year-by-year meteorological inputs. The PM$_{2.5}$
simulations from the *BASE* scenario also supported the PM$_{2.5}$ estimates in Sect. 2.1.

The evaluation of meteorological simulations of surface temperature, surface relative humidity, surface wind speed, and surface wind direction from WRF against ground-level observations from the National Climate Data Center (NCDC, ftp://ftp.ncdc.noaa.gov/pub/data/noaa/) were summarized in Fig B1. The WRF model well repoduced the near-surface temperature (r=0.98, normalized mean bias=-1.9%) and relative humidity (r=0.81, normalized mean bias=5.4%), but slightly
overestimated surface wind speed (r=0.57, normalized mean bias=8.0%). The WRF simulation quality of temperature, relative humidity and wind direction was consistent across years, but the simulation quality of wind speed showed slightly larger inter-annual variations. The validation results showed that the WRF simulations was acceptable to support further simulation of PM$_{2.5}$ concentrations. The evaluation of PM$_{2.5}$ simulations from CMAQ during the time period when ground measurements are available has been reported in our previous study (Zhang et al., 2019a). Compared to the measurements
from ground monitoring stations, our model simulations well reproduced the spatial and temporal distributions of PM$_{2.5}$ across China. Compared to the daily PM$_{2.5}$ measurements in 74 cities, the CMAQ simulations obtained correlation coefficient r higher than 0.6 in 67 cities. The simulated PM$_{2.5}$ decrease (30%) during 2013-2017 over China also well matched the observed PM$_{2.5}$ decrease (33%).

## 3. Results and Discussion

### 3.1 Evaluation of the two-stage PM$_{2.5}$ prediction model

The SMOTE resampling approach improved the prediction accuracy in the five-fold CV that the area under the curve (AUC) increased from 90.7 to 98.7 (Fig. B2). The two-stage model predictions in the five-fold CV matched the ground measurements



well with an $R^2$ of 0.80 and RMSE of 18.5 μg/m$^3$ (Fig. B2). The prediction accuracy in the by-location CV ($R^2$ of 0.71 and RMSE of 22.1 μg/m$^3$) and by-year CV ($R^2$ of 0.58 and RMSE of 27.5 μg/m$^3$) was lower than that in the five-fold CV, indicating unobserved temporal and spatial trends contributed to the PM$_{2.5}$ prediction. The model performance was comparable to that reported in previous studies (Xiao et al., 2018;He and Huang, 2018;Dong et al., 2020).

Specifically, compared to a benchmark model without SMOTE resampling and setting the PM$_{2.5}$ concentration as the dependent variable, the two-stage model in this study better predicted high-pollution events (Fig. 2). The density distribution of the PM$_{2.5}$ predictions from the two-stage model was very close to the density distribution of the PM$_{2.5}$ measurements. The density distribution of the PM$_{2.5}$ predictions from the benchmark model showed a higher percentage of low PM$_{2.5}$ concentrations and a lower percentage of high PM$_{2.5}$ concentrations than those revealed by the density distribution of the measurements. The greater ability of our two-stage model in capturing the daily variations in PM$_{2.5}$ concentrations could better support our following analysis about meteorological impacts.

### 3.2 Long-term trends of PM$_{2.5}$ concentrations over East China

Figure 3 shows the PM$_{2.5}$ trends during 2000–2018 in East China, as well as the key regions including the Beijing-Tianjin-Hebei (BTH) region, the Fen-wei Plain (FWP), the Yangtze River Delta (YRD) and the Pearl River Delta (PRD). The PM$_{2.5}$ concentrations continuously increased from 35.4 μg/m$^3$ in 2000 to 48.7 μg/m$^3$ in 2006 over East China. It then remained relatively constant from 2007–2013 and decreased from 46.5 μg/m$^3$ in 2013 to 32.5 μg/m$^3$ in 2018. BTH and FWP showed consistent temporal trends of PM$_{2.5}$, with higher pollution levels over BTH. However, the difference in PM$_{2.5}$ level between BTH and FWP has greatly decreased since 2015 due to the higher rate of PM$_{2.5}$ decrease in BTH resulting from the stricter emission control policies. The PM$_{2.5}$ level in the PRD reached its peak in 2006 and decreased thereafter. The observed PM$_{2.5}$ concentration in 2018 was 14.0, 30.9, 18.2, 22.9, and 13.2 μg/m$^3$ lower than that in 2013 over East China, BTH, FWP, YRD, and PRD, respectively.

### 3.3 Interannual and seasonal trends of meteorology-associate PM$_{2.5}$

Figure 4 shows the meteorological contribution in monthly average PM$_{2.5}$ concentrations estimated from the GAM model and CMAQ simulations. The temporal trends of meteorology-associate PM$_{2.5}$ estimated from these two methods were consistent across East China and in the key regions, with the correlation coefficients ranging between 0.53 (East China) and 0.72 (BTH). For example, the GAM model estimated typical favorable meteorological conditions in Oct 2013, Oct 2012 and Feb 2016 in BTH, which are also captured by the CMAQ model. However, the magnitude of the meteorological effects estimated by CMAQ were slight higher than GAM.

Figure 5 shows the GAM estimated temporal trend in meteorology-associate PM$_{2.5}$ across East China. Consistent with the CMAQ estimation (Fig. C1), 2012 is a typical year during which the meteorological conditions were favorable to PM$_{2.5}$ pollution control over East China, with an annual meteorology-associate PM$_{2.5}$ anomaly of -1.8 μg/m$^3$, 4.07%) (Fig. 5). 2004



is a typical year during which the meteorological conditions were unfavorable to PM$_{2.5}$ pollution control, with an annual

meteorology-associate PM$_{2.5}$ increase of 1.2 μg/m$^3$ (2.60%). The meteorological effects changed drastically over a relatively short time period. For example, in 2005, the meteorological conditions were greatly favorable to pollution control, but in the previous and following years, i.e., 2004 and 2006, respectively, the meteorological conditions were greatly unfavorable to pollution control. The long-term trend of the annual meteorology-associate PM$_{2.5}$ fluctuated about 0 across East China, with a decreasing trend (the meteorological conditions improving) from 2003–2010 and an increasing trend (the meteorological

conditions worsening) from 2010–2017 (Fig. 5, Fig. B3). The CMAQ simulations estimated the largest unfavourable meteorological contribution in 2018 of 11.0%, and the greatest benefical meteorological contribution in 2012 of 7.2% over East China.

The interannual variations in the meteorology-associate PM$_{2.5}$ assessed in this study were consistent with those reported in previous studies (Zhang et al., 2018). For example, Feng et al. (2020) presented the long-term variations in air stagnation in

north China that characterized the circulation and diffusion in the boundary layer with fixed emissions to describe the temporal trend of haze-related weather conditions. The temporal pattern of the air stagnation index from 2000–2018 was closely resembled the temporal trend of the estimated meteorological-associate PM$_{2.5}$ in this study. Additionally, we observed unfavorable meteorological conditions in the winters of 2014 and 2016, consistent with the previously reported climate anomalies in these two years (Yin et al., 2017;Yin and Wang, 2017). We also showed that the meteorological

conditions in 2014 and 2015 were more unfavorable to PM$_{2.5}$ pollution control than those in 2013 over East China, as previously reported (Zhang et al., 2019b;Wang et al., 2019a).

Since haze events that greatly affects public health mainly occur in fall and winter (Zhao et al., 2013), we further analyzed the meteorological effects during fall-winter (September, October, November, December, January, and February) and spring-summer. The meteorological conditions in fall-winter dominated the annual meteorological effects on PM$_{2.5}$. We observed

typical unfavorable meteorological conditions in the fall-winter of year 2006 (2.8 μg/m$^3$) and 2016 (2.5 μg/m$^3$). In certain years, e.g., 2018, the spring-summer meteorological conditions were unfavorable to pollution control, but since the fall-winter meteorological conditions were favorable, the annual meteorological effect was beneficial. The significant fall-winter meteorological effects indicated the critical contribution of meteorology to haze event formation. The fall-winter weather conditions in 2017 were substantially better than the fall-winter weather conditions in 2013, leading to a 3.3 μg/m$^3$ decrease

in the meteorology-associate PM$_{2.5}$, thereby contributed to the achievement of pollution control targets of the Action Plan (Zhang et al., 2019b;Yi et al., 2019). Since the current evaluation of clean air policies focuses on changes in pollution levels over short periods, e.g., three or five years, policy performance can be largely affected by meteorological changes.

### 3.4 Spatial heterogeneity in meteorology-associate PM$_{2.5}$ trends

We also analyzed the variations in the meteorological influence on PM$_{2.5}$ in several populous urban agglomeration regions of

250 China (Fig. 5, Fig. B3). In the BTH region, 2014 was a typically unfavorable year (3.1 μg/m$^3$), and 2010 was a typically



favorable year (-4.9 µg/m³). The shape of the interannual trend of the meteorology-associate PM$_{2.5}$ during wintertime in BTH was consistent with that in previous studies. For example, the 2014 and 2017 winter meteorological conditions were greatly favourable and the 2016 winter meteorological conditions were considerably unfavorable (Yi et al., 2019;Wang and Zhang, 2020). The meteorological effects showed a regional consistency with varying magnitudes. For example, 2004 was a typical

unfavorable year in both the PRD (6.3 µg/m³) and the YRD (2.7 µg/m³), and 2016 was a typical favorable year in both the PRD (-7.3 µg/m³) and the YRD (-2.1 µg/m³). Consistent with previous studies, the PRD revealed the largest meteorological influence on PM$_{2.5}$ among these regions (Zhai et al., 2019).

We observed notable regional heterogeneity in the long-term trends as well as seasonal trends of the meteorological effects on PM$_{2.5}$ (Fig. 5, Fig. B3). In the northern part of China, especially in the North China Plain and central East China, the

260 meteorological conditions worsened and were adverse to pollution control (Yin and Wang, 2018;Zhang et al., 2018). Multiple climate systems could be associated with the long-term trend of meteorological effects. For example, greenhouse gas-induced warming may result in a decrease in light-precipitation days and surface wind speed, which are unfavorable to pollution control (Chen et al., 2019). In contrast, in the southern part of China, especially in the YRD and surrounding regions, the estimated meteorological conditions were improving and were beneficial to pollution control (Chen et al., 2019).

Regarding the seasonal trend of the meteorological effects, in spring-summer, we observed improving meteorological effects in the southern part of China and worsening meteorological effects in the northern part of China. This spatially heterogeneous trend may result from the strengthening of the East Asia summer monsoon, which enhances the transportation of aerosols from the south to the north of China (Zhu et al., 2012;Liu et al., 2017a). In fall-winter, the East Asia winter monsoon significantly affects air pollution levels that benefits the air quality in North China but is unfavourable to air quality

in the South China due to the southward transport of pollutant from north to south (Jeong and Park, 2017;Yin et al., 2015). For example, in the year 2004, 2005, 2007, and 2010 with strong East Asia winter monsoon, the BTH and the FWP showed strong favourable meteorology contributions to PM$_{2.5}$, but the YRD and the PRD showed unfavourable meteorological effects. On the contrary, in the year 2006 with weak East Asia winter monsoon, the BTH and the FWP showed unfavourable meteorological effects (Jeong and Park, 2017).

The large-scale atmospheric circulations in some specific years also showed notably distinct effects on PM$_{2.5}$ concentrations over the north and south of East China, due to the opposite effects on meteorology parameters. For example, in 2015 and 2016 with strong El Niño, the fall-winter meteorology in the northern part of East China was significantly unfavorable for pollution control but that in the southern part of East China was considerably favorable. One reason is that the El Niño leads to excessive precipitation over southern China that in favour of wet deposition, but weakened the East Asia winter monsoon

and led to south wind anomaly, weaker surface wind, and high humidity that were favorable to pollution events in the northern region of East China (Yin et al., 2015;Yin and Wang, 2016;He et al., 2019;Chang et al., 2016). On the country, during the year with La Niña , e.g. 2007 and 2010, we estimated beneficial winter-fall meteorology in northern regions but unfavourable meteorology in the southern region (Cheng et al., 2019b).



Consistent with previous studies, we also observed spatially varying associations between PM$_{2.5}$ and meteorological
parameters that reflect the varying PM$_{2.5}$ responses to meteorological changes (Fig. B4). Temperature was positively associated
with PM$_{2.5}$ in spring, summer and fall across East China; however, in winter, the temperature was negatively associated with
PM$_{2.5}$ in northern China (He and Wang, 2017;Qiu et al., 2015) due to the low-temperature-related stable atmosphere and
decreased evaporation loss of PM$_{2.5}$. Humidity yielded positive effects in northern China and negative effects in southern China
in all seasons, especially in winter (He et al., 2017). The spatial difference in the effects of humidity on PM$_{2.5}$ may occur due
to a threshold of the humidity altering the direction of the humidity influence, from hygroscopic increase to wet deposition.
The boundary height and precipitation were negatively associated with PM$_{2.5}$ across East China in all seasons, and the effect
of precipitation was greater in northern China than that in southern China (Wang and Chen, 2016). Regarding the relative
contribution of the different meteorology parameters, we found that over the south coast region, temperature and humidity
showed greater effects than did the boundary layer height and precipitation. In winter, humidity, boundary layer height and
precipitation were critical for the PM$_{2.5}$ variations in the middle and north of China. In summer and fall, the temperature and
humidity were critical for the PM$_{2.5}$ variations across southern China. In spring, the temperature showed notable effects in the
south coast region, and the precipitation exhibited large effects in the North China Plain.

### 3.5 PM$_{2.5}$ trends after adjusting the meteorological effects

In East China, after adjusting for the meteorological influence, PM$_{2.5}$ started increasing in 2000 and peaked in 2006 with an
increase of 9.6 µg/m$^3$ compared to the 2000 level (Fig. 6). Then, the PM$_{2.5}$ varied, with the second highest PM$_{2.5}$ level
occurring in 2011 (9.4 µg/m$^3$ higher than the 2000 level). After 2013, with the implementation of aggressive emission control
policies, PM$_{2.5}$ notably decreased, with a 13.1 µg/m$^3$ lower PM$_{2.5}$ level in 2018 compared to the level in 2013. After adjusting
for the meteorological effects, the temporal variations in PM$_{2.5}$ were consistent with the temporal variations in pollutant
emissions retrieved from the MEIC emissions. The emissions of SO$_2$ and PM$_{2.5}$ peaked in 2006, and the emissions of NO$_x$
peaked in 2012.

In the BTH region, PM$_{2.5}$ peaked in 2006 and decreased by 10.8 µg/m$^3$ in 2008 due to the emission control policies targeting
the air quality during 2008 Beijing Olympic Games. After 2008, PM$_{2.5}$ continuously increased and peaked in 2013, at an
increase rate of 1.0 µg/m$^3$ per year. Considering the variations in pollutant emissions, the first PM$_{2.5}$ peak in BTH was
primarily driven by SO$_2$ emissions, and the second PM$_{2.5}$ peak was driven by NO$_2$ and PM$_{2.5}$ emissions. The PM$_{2.5}$
decreasing trend after 2013 in BTH was higher than that in the other regions (5.8 µg/m$^3$ per year), mainly driven by the
emission reduction in SO$_2$ and PM$_{2.5}$. The annual average meteorology-adjusted PM$_{2.5}$ concentration in BTH from 2014-
2018 was consistent with that in a previous study (Qu et al., 2020). We found that the observed high-pollution events in the
fall-winter of year 2006, 2013, and 2016 were partly attributable to unfavorable meteorological conditions that led to a 5.9,
3.4, and 11.1 µg/m$^3$ PM$_{2.5}$ increase, respectively. Since the meteorology contributed up to 25% of the observed PM$_{2.5}$ level in
fall-winter, further emission control measures are needed to improve the winter air quality and avoid violations of the air



quality standards under unfavorable meteorological conditions. In FWP, the highest $PM_{2.5}$ level occurred in 2005, and the average decrease rate after 2013 was 2.8 µg/m$^3$ per year. The high pollution in the 2016 fall-winter period attributable to unfavorable meteorological conditions was also observed in FWP, although the meteorological effects in FWP were smaller than those in the BTH region, with up to 10% of the meteorology contribution in $PM_{2.5}$ in fall-winter. In the YRD, $PM_{2.5}$

peaked in 2011 and 2015. The unfavorable meteorological conditions observed in the fall-winter of 2016 did not occur in either the YRD or the PRD, showing a spatial difference in the meteorological system. In the PRD, $PM_{2.5}$ peaked in 2006 and continuously decreased from 2006-2018, at an average decrease rate of 2.8 µg/m$^3$ per year. This decreasing trend was consistent with the trend of the $PM_{2.5}$ emissions. The temporal variations in $NO_x$ and $SO_2$ emissions contributed to the trends in the meteorology-adjusted $PM_{2.5}$ from 2010–2011.

It is observed that although emissions dominated the interannual variations in $PM_{2.5}$, meteorological conditions significantly affected the observed $PM_{2.5}$ concentration in all key regions, especially in fall and winter. We observed as much as 25.6%, 6.3%, 5.1% and 4.8% annual average meteorological effects, estimated from GAM, in the PRD, BTH, FWP, and YRD, respectively, during the study period. The meteorological contributions in fall-winter were even higher. The CMAQ simulations estimated as much as 17.5%, 8.8%, 26.6%, and 6.6% annual average meteorological effects in the PRD, BTH,

FWP, and YRD, respectively. From 2015 to 2016, the winter-fall meteorological conditions considerably changed to unfavorable for pollution control in North China, leading to a 2.8 µg/m$^3$ increase in the winter-fall average $PM_{2.5}$ concentration across East China. BTH and FWP showed a 9.8 and 8.1 µg/m$^3$ increase, respectively. Such an increase may weaken the effects of emission control policies during this period. In 2018, the $PM_{2.5}$ concentration in Beijing was reported to be 51 µg/m$^3$. However, if 2018 had been a typical year with unfavorable meteorological conditions, the annual $PM_{2.5}$ concentration could

have reached 54 µg/m$^3$.

The meteorology-adjusted $PM_{2.5}$ trend from 2013-2018 showed varying spatial patterns. The highest decrease occurred in Beijing, Tianjin, south of Hebei and the capital cities, including Xi'an, Wuhan, Zhengzhou, and Changsha (Fig. 7), indicating the more efficient implementation of clean air policies in these regions. As described above, the effects of meteorology also showed spatial differences. Over the Northeast China Plain, North China Plain, and Sichuan Basin, the adjusted $PM_{2.5}$

decreasing trend was weaker than the observed trend. Over the Shanxi, the intersection of Hubei-Henan-Anhui and south of Jiangsu, the adjusted $PM_{2.5}$ decreasing trend was stronger than the observed trend. The interquartile range of the meteorological effects on the $PM_{2.5}$ trend varied between -17.2% and 1.8% across East China. From 2013-2018, the decreasing trend of the meteorology-adjusted $PM_{2.5}$ level was lower than that of the observed $PM_{2.5}$ level by 8.4% in East China, 7.9% in the BTH region, 3.3% in the YRD, and 7.5% in the PRD while the adjusted trend was greater than the

observed trend by 2.01% in the FWP.


### 3.6 Sensitivity analysis

To evaluate whether the selection of statistical models affects the assessed associations between meteorology and $PM_{2.5}$, we compared the meteorology-associate $PM_{2.5}$ estimated by GAM and MLR. The estimated meteorology-associate $PM_{2.5}$ levels from the MLR and GAM matched well, with correlation coefficients larger than 0.98 across East China (Fig. B5). Hence, the results of this study are robust and not affected by the selection of $PM_{2.5}$-meteorology model.

To examine the effects of length of the time window when constructing the deseasonalized $PM_{2.5}$, we conducted a sensitivity analysis with a 90-day averaging window in the BTH region, and the estimated $PM_{2.5}$ concentrations after adjusting for meteorological effects were almost identical to the results using a 50-day time window (Fig. B5). Thus, this statistical method was not sensitive to the averaging time window.

Compared to previous studies, we employed the GAM to better describe the nonlinear associations between $PM_{2.5}$ and meteorology in this study. We observed consistent temporal trends of the meteorological effects and the meteorologically adjusted $PM_{2.5}$ concentrations compared to previous studies, but the magnitude of the assessed meteorological effects and adjusted $PM_{2.5}$ concentrations varied. Thus, when comparing the meteorological effects of a specific year, the conclusion may be inconsistent (Xu et al., 2020;Zhai et al., 2019;Zhang et al., 2019a;Zhang et al., 2019b). Assessing the meteorology-associate $PM_{2.5}$ with different methods may also lead to varying long-term trends (Xu et al., 2020). Several factors may affect the uncertainty of the assessed meteorological contributions in this study. First, the satellite retrievals exhibited an increasing prediction error when hindcasting historical pollution levels. The satellite-driven $PM_{2.5}$ prediction model used in this study is a state-of-the-art prediction model with improved prediction accuracy for high-pollution events, but the hindcast prediction quality needs to be further improved to better describe the historical $PM_{2.5}$ spatiotemporal distribution. Second, we obtained meteorological information from the MERRA-2 reanalysis dataset with a spatial resolution lower than that of the $PM_{2.5}$ predictions. This resolution mismatch with smooth spatial variations in the meteorological fields may not fully describe the meteorological effects at the local scale.

### 4. Conclusions

In this study, we analyzed the meteorology- and emission-driven variations in the $PM_{2.5}$ concentration during 2000-2018 across East China by the GAM-based method and CMAQ simulations. To support the GAM-based analysis, we combined satellite data, CMAQ simulations and ground observations to predict complete-coverage $PM_{2.5}$ concentrations with a two-stage machine learning model that attained improved prediction accuracy of high-pollution events. Both methods showed significant meteorological influences on $PM_{2.5}$ dominated by the meteorological conditions in fall and winter. The greatly varying meteorological effects on $PM_{2.5}$ concentration over a relatively short time period may remarkably affect the evaluation of clean air policies during a certain period. We also observed distinct regional differences in the long-term and seasonal trends of the meteorological effects. The meteorology-associate $PM_{2.5}$ tended to increase in the North China Plain and Central China, but





decrease across southern China, e.g. in the YRD. After adjusting for the meteorological effects, the average PM$_{2.5}$ concentration decreased 13.1 μg/m$^3$ from 2013–2018 over East China, and the BTH region showed the greatest decrease (28.5 μg/m$^3$) among the studied urban agglomeration regions. The decreasing trend of PM$_{2.5}$ after adjusting for the meteorological effects was 8.4% weak than the observed PM$_{2.5}$ decreasing trend in East China, 7.9% weak in the BTH region, 3.3% weak in the YRD, and 7.5% weak in the PRD while the adjusted trend was 2.0% greater than the observed trend in the FWP. Considering the remarkable meteorological contributions to PM$_{2.5}$, further emission reduction measures are required to prevent the occurrence of haze events under unfavourable meteorological conditions.

**Appendix A. Data collection and processing**

We collected hourly PM$_{2.5}$ measurements from 2013-2018 from both the national air quality monitoring network (~1,593 stations) and local air quality monitoring stations (~ 1,700 stations) mainly located in East China. Continuous identical measurements over at least three hours were removed due to instrument malfunction. Daily average concentrations were calculated based on at least 12 hourly measurements.

We obtained Aqua and Terra MODIS Collection 6 level 2 aerosol products at a 0.1-degree resolution from https://ladsweb.modaps.eosdis.nasa.gov/. Since the aerosol optical depth (AOD) retrieved with the Deep Blue (DB) algorithm and the Dart Target (DT) algorithm (Levy et al., 2013;Hsu et al., 2013) exhibit different coverage and retrieval accuracy (Wang et al., 2019b), we fitted daily linear regressions to fill the missing retrievals when only DT or DB AOD was presented. Then, we calculated the average of the DT AOD and DB AOD separately for each sensor. Similarly, since the Aqua AOD and Terra AOD are observed at different pass over times, to improve the data coverage, we fitted daily linear regressions to fill the missing retrievals when only Aqua AOD or Terra AOD was presented. We calculated the average of the Aqua and Terra AODs to characterize the daily aerosol loadings (Jinnagara Puttaswamy et al., 2014).

We also used daily PM$_{2.5}$ simulations at a spatial resolution of 36 km during 2000-2018 from the WRF/CMAQ model as an important predictor. The inverse distance weighting (IDW) method was applied to interpolate the CMAQ simulations to match the grid of 0.1°. Detailed description of the WRF/CMAQ simulations could be found in Sect. 2.3.

Meteorological parameters were extracted from the Modern-Era Retrospective analysis for Research and Applications Version 2 (MERRA-2) dataset at a resolution of 0.5° latitude × 0.625° longitude (Randles et al., 2017). We extracted parameters including surface albedo, cloud area fraction for low clouds, total cloud area fraction, surface net downward longwave flux, surface incoming shortwave flux, surface net downward shortwave flux, total incoming shortwave flux, total net downward shortwave flux, surface pressure, 2-meter specific humidity, 2-meter air temperature, 2-m dew point temperature, total column ozone, total column odd oxygen, total precipitable ice water, total precipitable liquid water, total precipitable water vapor, 2-meter eastward wind (U wind), 2-meter northward wind (V wind), 10-meter U wind, 10-meter wind speed, 10-meter V wind, U wind at 500 hPa, U wind at 850 hPa, V wind at 500 hPa, V wind at 850 hPa, total latent



energy flux, evaporation from turbulence, planetary boundary layer height, snowfall, and bias-corrected total precipitation. These parameters have been reported to be strongly associated with the $PM_{2.5}$ concentration in various regions in China

(Chen et al., 2020). The inverse distance weighting method was applied to estimate the daily smooth surface of meteorological data and to match with the modelling grid at a 0.1° spatial resolution.

Elevation data from the Global Digital Elevation Model (GDEM, https://earthexplorer.usgs.gov/) version 2 at a 30-m resolution were averaged to match the modelling grid. We calibrated the gridded population distribution data from the LandScan Global Population Database (https://landscan.ornl.gov/), the Gridded Population of the World (GPW,

https://beta.sedac.ciesin.columbia.edu/data/set/gpw-v4-population-count) dataset and the WorldPop dataset (https://www.worldpop.org/) at the county level with the total population reported in China City Yearbooks. These calibrated gridded population data were fused to better characterize the population distribution across China (Xiao et al., 2020).The land cover classification data of urban and rural regions at a 30-m resolution for 2000-2017 were downloaded from http://data.ess.tsinghua.edu.cn (Gong et al., 2019a;Gong et al., 2019b). The fraction of urban/rural region at the 30-m resolution

was averaged according to the modelling grid.





## Appendix B. Model performance evaluation

The hyperparameters of XGBoost, including the maximum number of boosting iterations, the learning rate, the maximum depth of a tree, the minimum sum of the instance weight needed in a child, the subsampling ratio of a training instance, and the subsampling ratio of columns when constructing each tree, were optimized by grid search with the five-fold cross-validation (CV) root-mean-square error (RMSE) as a performance evaluation statistic.

The model performance was evaluated through five-fold CV, by-year CV and by-location CV. The five-fold CV approach randomly selects 20% of the data for model testing and train the model with the remaining data. This process is repeated five times, and each record is selected once as testing data. The by-year CV approach validates the model hindcast ability by sequentially selecting one year of data for testing and using the remaining yearly data for model training such that each year is selected once for testing. The by-location CV approach validates the model ability for spatial prediction by using the data at 20% randomly selected locations for testing and uses the remaining data for model training. This process is repeated five times until each location has been selected once for model testing.



**Table B1. Model fitting average $R^2$ value of the seasonal generalized additive model (GAM), year-round GAM, seasonal stepwise multiple linear regression (MLR), and year-round MLR.**

|  | Spring | Summer | Fall | Winter | Year-round |
|---|---|---|---|---|---|
| Seasonal GAM | 0.39 | 0.45 | 0.42 | 0.48 | |
| Year-round GAM | | | | | 0.32 |
| Seasonal MLR | 0.34 | 0.40 | 0.37 | 0.42 | |
| Year-round MLR | | | | | 0.26 |



**Figure B1: Evaluation of the WRF model simulations. The correlation coefficient and normalized mean bias was**
445 **calculated by comparing WRF simulations with ground observations from the National Climate Data Center.**



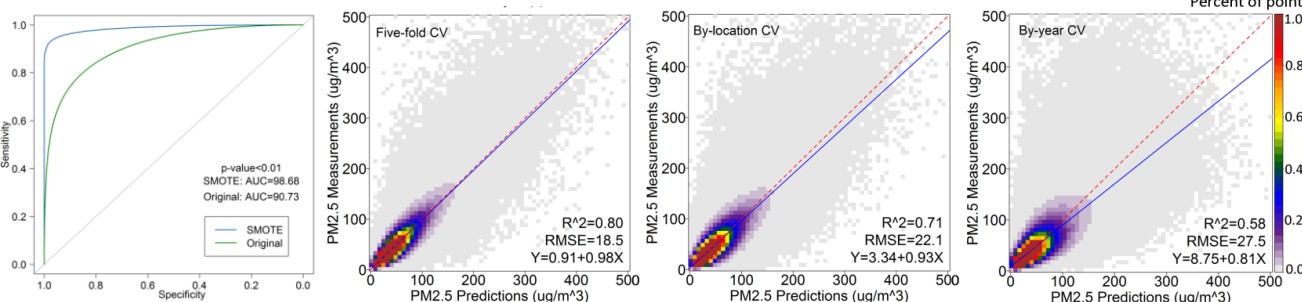

**Figure B2: Model evaluation of the first-stage model trained with the original dataset and the SMOTE-resampled dataset in five-fold cross-validation (CV) and scatter plots comparing the ground measurements and model**

**predictions in five-fold CV, by-location CV and by-year CV.**


**Appendix C. Meteorological contributions to PM2.5.**

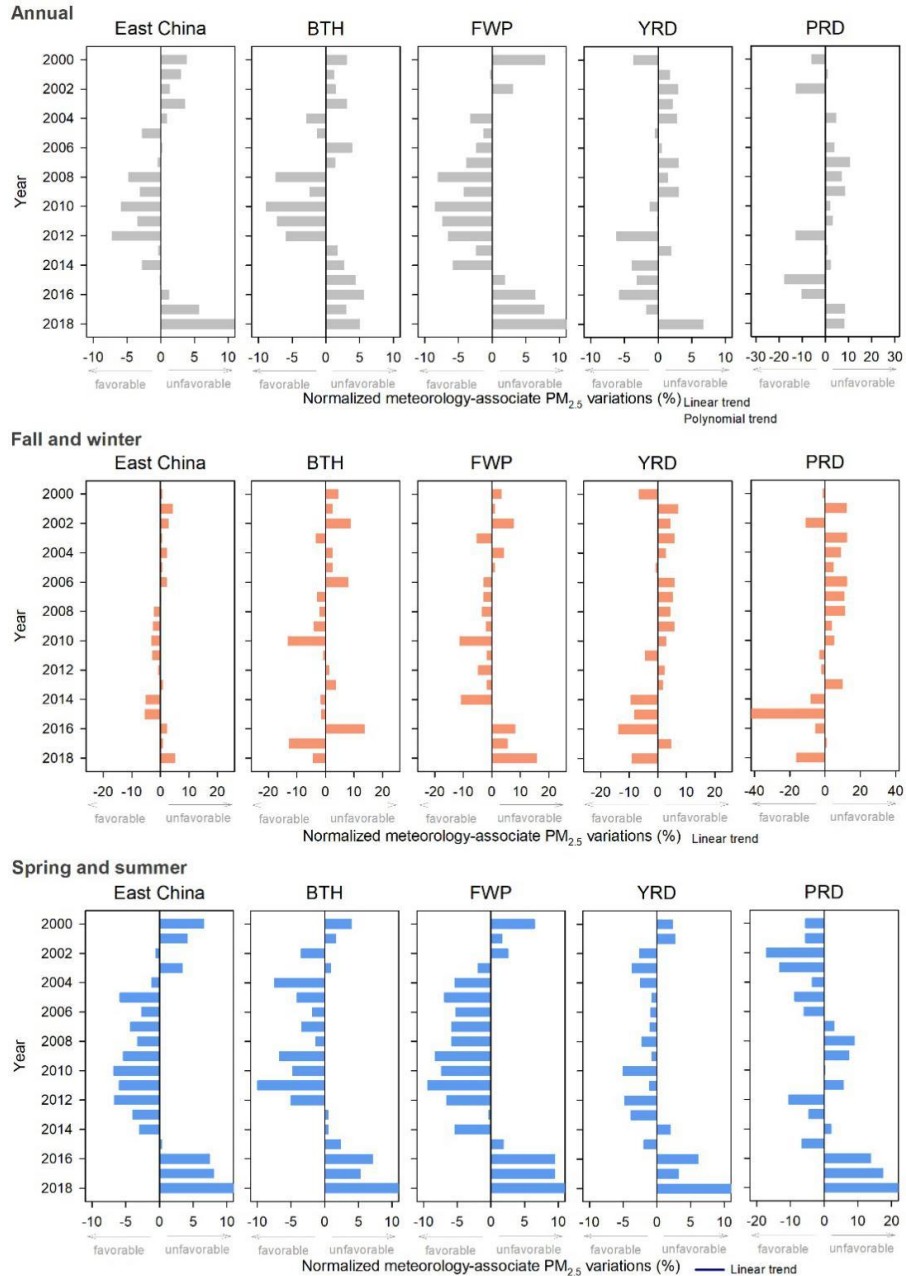

**Figure C1: The CMAQ estimated relative impact of meteorology on annual average PM2.5 (top row), relative impact of meteorology on average PM2.5 in fall-winter (September, October, November, December, January in next year, and February in next year) (middle), and relative impact of meteorology on average PM2.5 in spring-summer (bottom row) with the long-term trends estimated by polynomial and linear regression over East China, BTH, FWP, YRD, and PRD.**



**Figure C2: Distribution of the estimated seasonal coefficients of the normalized meteorological parameters in East China.**



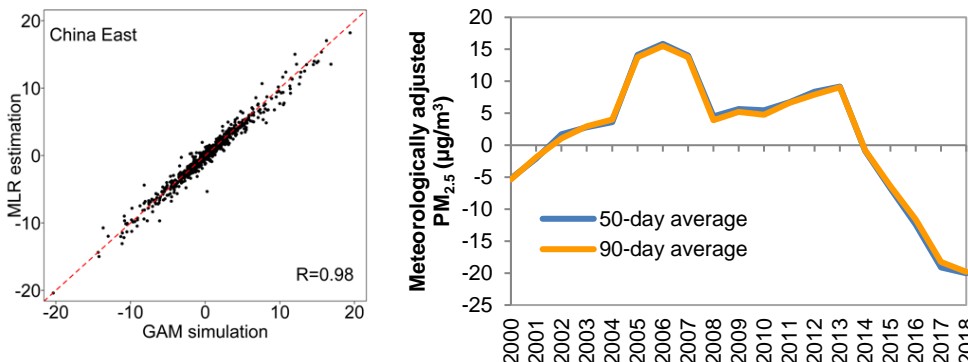

**Figure C3: Meteorology-associate PM$_{2.5}$ variations estimated with the MLR and GAM (left), and meteorologically adjusted PM$_{2.5}$ with 50-day seasonal averaging window and 90-day seasonal averaging window (right).**

### Data availability

All the data used to predict PM$_{2.5}$ concentrations are openly available for download from the websites given in the supplement.

### Author contributions

Qingyang Xiao, Guannan Geng, and Qiang Zhang designed the analyses and Qingyang Xiao carried them out. Yixuan Zheng performed the WRF/CMAQ simulations. Xiaomeng Huang optimized the data fusion model. Cuihong Chen, Huizheng Che, Xiaoye Zhang, and Kebin He interpreted the results. Qingyang Xiao and Guannan Geng prepared the manuscript and figures with contributions from all co-authors.

### Competing interests

The authors declare that they have no conflict of interest.

### Acknowledgments

This work was supported by the National Natural Science Foundation of China (grant no. 42007189, 42005135, 41921005, and 41625020).



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

645





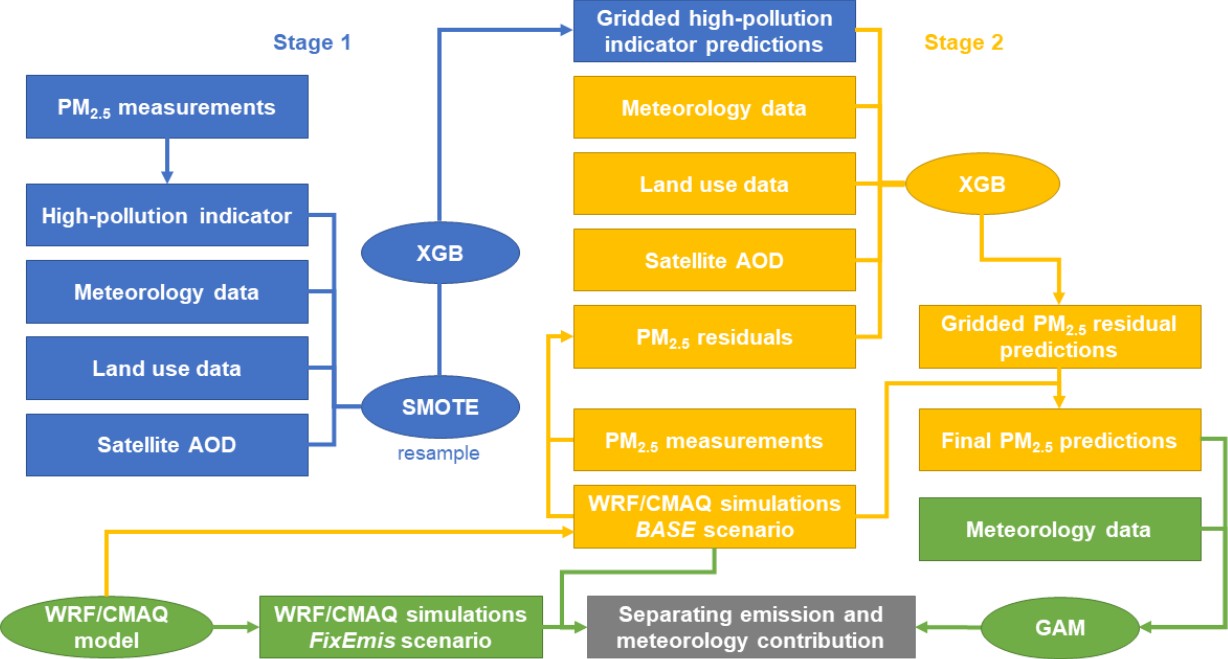

**Figure 1: Methodology framework of this study.**





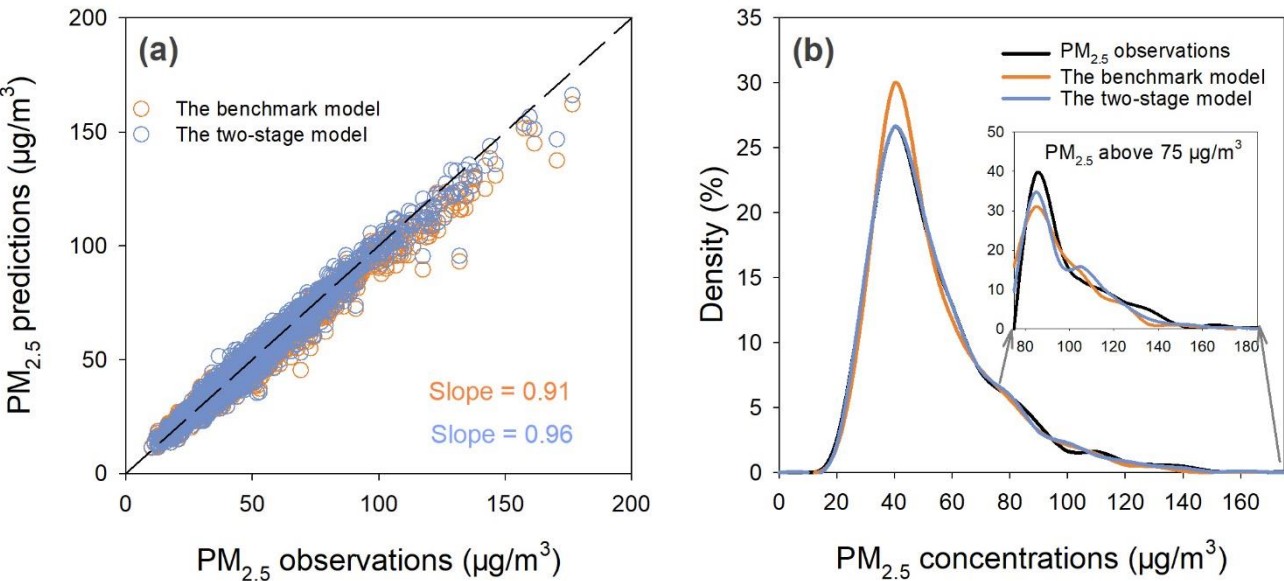

**Figure 2: Comparisons between the two-stage model and the benchmark model. (a) The scatter plot of the two-stage model predictions and the benchmark model predictions against ground observations in the five-fold cross-validation (CV). (b) Density distributions of the two-stage model predictions, the benchmark model predictions and the PM$_{2.5}$ observations in the five-fold CV.**




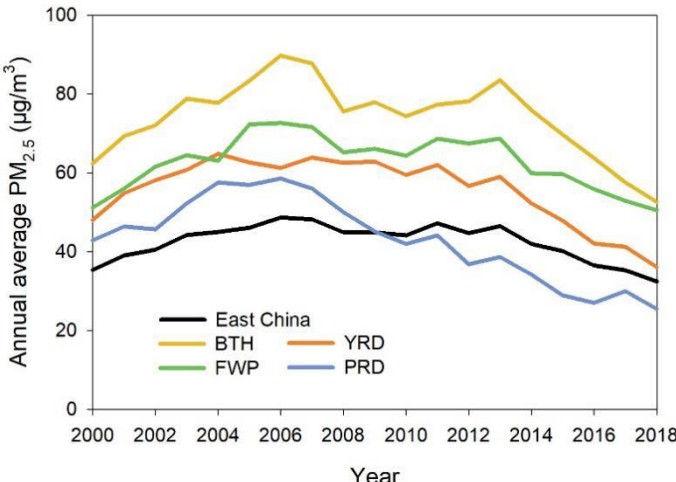

**Figure 3: Temporal trends of the annual average satellite-based PM₂.₅ concentrations over East China and the key regions during 2000–2018. BTH: Beijing-Tianjin-Hebei; FWP: Fen-wei Plain; YRD: Yangtze River Delta; PRD: Pearl River Delta.**



**Figure 4: Fractional contribution of meteorology to PM₂.₅ concentrations on the monthly scale during 2000–2018, estimated from CMAQ (the orange line) and GAM (the blue line).**

**Figure 5: The GAM estimated relative impact of meteorology on annual average PM$_{2.5}$ (top row), relative impact of meteorology on average PM$_{2.5}$ in fall-winter (September, October, November, December, January in next year, and February in next year) (middle), and relative impact of meteorology on average PM$_{2.5}$ in spring-summer (bottom row) with the long-term trends estimated by polynomial and linear regression over East China, BTH, FWP, YRD, and PRD.**





**Figure 6: Time series of the annual average (left column) and fall-winter average (middle column) PM₂.₅ concentrations before (the orange line) and after (the gray line) the adjustment of the meteorological effects from 2000–2018 using GAM. The gray shadow shows the potential range of the observed PM₂.₅ due to meteorological effects. The right column shows the MEIC emissions of PM₂.₅, NOₓ, and SO₂, respectively.**

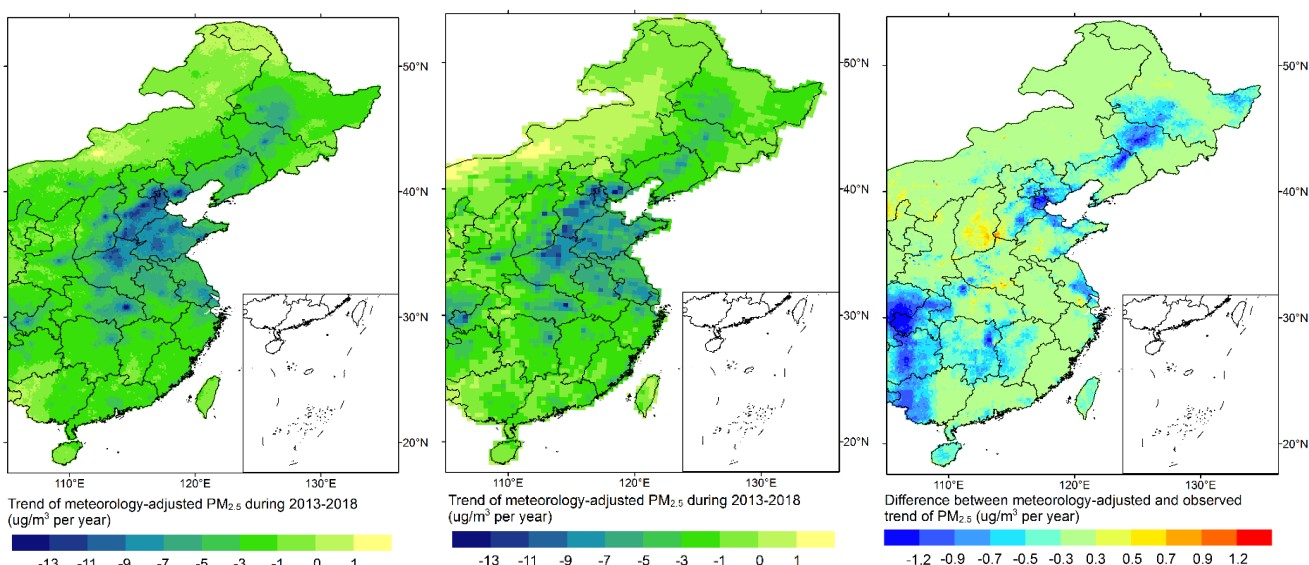

**Figure 7: Left: Spatial distribution of the PM₂.₅ decrease rate during 2013–2018 after adjusting for the meteorological effects using GAM. Middle: Spatial distribution of the PM₂.₅ decrease rate during 2013–2018 after adjusting for the meteorological effects using CMAQ. Right: The difference in the PM₂.₅ decrease rate before and after the adjustment for the meteorological effects using GAM.**