# Peer review of "Separating emission and meteorological contribution to long-term PM2.5 trends over East China during 2000–2018"

_Atmospheric Chemistry and Physics, 2021_

## Author Response (AR1)

Although fine particulate matter in the air originates from emissions, its concentrations are also affected by meteorology. Quantifying the impacts of meteorological conditions on air quality is critical for both understanding pollution process and air pollution policy. In this article, authors used a two-stage prediction model to estimate PM2.5 over China based on various data, and assessed meteorological effects on PM2.5 using a generalized additive model and WRF/CMAQ model. I am impressed by the workload. The writing, organization, and contents of this manuscript are of significant quality. I recommend this manuscript be accepted for publication after the following minor concerns are cleared.

**Response: We thank the referee for the encouragement and the valuable suggestions to improve our manuscript.**

1.   In the introduction section, it is highly recommended that the authors provide more discussion on the related previous studies about meteorological contribution to PM2.5 based on modeling sensitivity studies and statistical methods.

**Response: We added the following summary of previous studies in line 48-59 "Previous studies reported that the $PM_{2.5}$ and other air pollutants emissions dropped significantly during this period (Bian et al., 2019;Liu et al., 2015). Previous studies also estimated the contribution of meteorology to the air quality improvement. Zhang et al. (2019b) reported that about 13% and 20% of total $PM_{2.5}$ decline during 2013-2017 are due to meteorological effects in Beijing-Tianjin-Hebei (BTH) and Yangtze River Delta (YRD), respectively, estimated from the pollution-linked meteorological index (PLAM). Zhang et al. (2019a) reported that meteorological changes led to a 16% decrease and a 4% increase in $PM_{2.5}$ changes during 2013-2017 in BTH and YRD, respectively, estimated from a chemical transport model (CTM) simulations. Zhai et al. (2019) reported that after adjustment of meteorological effects, the $PM_{2.5}$ decline during 2013-2018 was 14% weaker in BTH and 3% stronger in YRD, respectively, estimated from a statistical model. Previous studies further analyzed the long-term trend of effects of meteorological systems and climate change on $PM_{2.5}$ pollution, especially in the context of global warming (Ruijin et al., 2017;Wang and Chen, 2016;Yi et al., 2019). For example, Feng et al. (2020) reported a trend of negative meteorological effects on air quality improvements in North China during 1980-2018, but the effects dropped during 2013-2018. Xu et al. (2020) also reported a trend of negative meteorological effects during 2000-2017 in Beijing, but an overall trend of beneficial meteorological effects in some provinces in North China." We also added more discussion on the uncertainties of chemical transport models and statistical methods in line 68-70 ("Uncertainties in the historical emission inventory as well as in the simulated meteorological fields affected the modelling results. Researchers' selection of chemical reaction mechanisms as well as parameter optimization could also lead to varying results (Chen et al., 2020).") and in line 76-78 ("Since the linear model may not fully characterize the non-linear associations and interactions between air pollution and meteorology, some studies also employed machine learning algorithms that better describe the complex relationships between meteorology and air pollution (Grange et al., 2018;Vu et al., 2019;Zhang et al., 2020;Qu et al., 2020).")**

2. Please clarify how to adjust the meteorological effects using GAM and CMAQ in methods.

**Response: We added the following sentences in line 164-166 "Thus, the meteorological effects on PM$_{2.5}$ long-term variations were assessed as the GAM-estimated responses of PM$_{2.5}$ to variations in the deseasonalized meteorological parameters." and in line 179-182 "The simulations of the *FixEmis* scenario were calibrated by multiplying the ratio between the satellitia-based PM$_{2.5}$ estimates in Sect. 2.1 and the *BASE* scenario PM$_{2.5}$ simulations. The meteorological effects on PM$_{2.5}$ long-term trends were assessed as the 10-day average of daily simulations in the *FixEmis* scenario minus the 19-year average of *FixEmis* simulations."**

3. Line 226, "benefical" should be "beneficial"

**Response: We corrected this word as suggested.**

4. Please modify the text above the color bar in Figure 7 with "GAM","CMAQ".

**Response: Thank you for your suggestion. We adjusted the Figure legend title as suggested.**

This paper separates the contributions of emissions and meteorology to PM2.5 trends in different regions of China for 2000-2018 by reconstructing the PM2.5 record back to 2000 using satellite AOD data and a machine-learning approach including information from PM$_{2.5}$ data,WRF-CMAQ model results, and meteorological variables. This is a remarkable piece of work and the first such analysis to my knowledge that goes back to 2000, thus providing a long-term perspective on the role of meteorology and enabling a better understanding of the relation of PM2.5 trends to anthropogenic emissions.  This includes better definition of the 2000s maximum. The analysis seems carefully done and the interpretation is insightful. A particularly interesting result, as the authors point out, is that interannual meteorological variability can play an important role in driving PM2.5 trends over the 3-5 year horizon of government action plans. I support publication but suggest a few editorial revisions.

**Response: We thank the referee for the positive tone and the valuable suggestions to improve our manuscript.**

1.    The writing is in general very good but there are recurring problems with the tense form of verbs. For example, the last sentence of the abstract should read 'is severe…is clustered'.   At various points in the paper, 'meteorology-associate' should be 'meteorology-associated'. The authors should check throughout.

**Response: We reviewed and corrected the verbs and other grammar errors.**

2.    I found Figure 1 to be incomprehensible and Section 2.1 riddled with machine-learning jargon, and this initially discouraged me from the paper. One way to fix Figure 1 would be with a detailed caption describing the different elements of the Figure. Section 2.1.2 and other portions of the text should be edited for a readership not steeped in machine-learning packages.

**Response: We added the following explanation in the caption of Figure 1,"The green process shows the two methods that separating emission and meteorology contributions to PM$_{2.5}$ in this study. The first method assesses the meteorology-associated PM$_{2.5}$ from WRF/CMAQ simulations with the fixed emissions at the 2000 level and varying meteorological inputs. The second method assesses the meteorology-associated PM$_{2.5}$ with satellite-based PM$_{2.5}$ estimations and a generalized additive model (GAM). The processing of satellite-based PM$_{2.5}$ estimation includes two stages. In stage 1 (blue), we constructed a measurement-based high-pollution indicator and trained an extreme gradient boosting (XGB) model to predict the high-pollution indicator. In stage 2 (yellow), we trained a XGB model to predict the residuals of WRF/CMAQ simulations with high-pollution indicator as well as satellite AOD, meteorology and land use data as predictors."**

**We also edited section 2.1.2 to make it easy to follow: "A two-stage prediction model was developed to estimate PM$_{2.5}$ concentrations over China (Fig. 1). The first-stage model described high-pollution events that were underestimated in previous models and the second-stage model predicted residuals of CMAQ PM$_{2.5}$ simulations with the estimated high-pollution indicator from the first-stage model.**

Since high-pollution events relatively rarely occur in the model training dataset, models may not appropriately characterize the associations between high $PM_{2.5}$ concentrations and predictors, leading to underestimation of high-pollution levels (Wei et al., 2020). We first defined a high-pollution indicator, describing whether the daily $PM_{2.5}$ observation was higher than the monthly average $PM_{2.5}$ concentration plus two standard deviations at each location. We noticed that only 3.9% of the daily data were assigned as high-pollution. To balance high-pollution samples and normal samples, the synthetic minority oversampling technique (SMOTE) (Torgo, 2010) that improved classifiers' performance in previous studies (Ghorbani and Ghousi, 2020;Saputra and Suharjito, 2019) was applied. The SMOTE algorithm oversampled the high-pollution data (the minority) by artificially generated new synthetic samples along the line between the high-pollution data and their selected nearest neighbors (Chawla et al., 2002;Chawla et al., 2003). This method also under-sampled the normal data (the majority) to better balance the model training dataset. After SMOTE resampling, high-pollution data accounted for 23% in the new model training dataset.

The balanced model training dataset was adopted to train the first-stage extreme gradient boosting (XGBoost) model that built the relationship between the high-pollution indicator and all the predictors, excluding CMAQ simulations. The predicted high-pollution indicator from the first-stage model was passed to the second-stage model as a predictor. We adopted the residual between the $PM_{2.5}$ measurement and the CMAQ $PM_{2.5}$ simulation as the dependent variable to train the second-stage model, thus enhances the response of predictors to $PM_{2.5}$ variations and improved the prediction accuracy."

3. Lines 260-263: it would be worth citing other papers that projected the effect of climate change in BTH, particularly since they did not agree: (1) Cai, W., Li, K., Liao, H., Wang, H., and Wu, L.: Weather conditions conducive to Beijing severe haze more frequent under climate change, Nat. Clim. Change, 7, 257–262, 2017; (2) Shen, L., D.J. Jacob, L.J. Mickley, Y. Wang, and Q. Zhang, Insignificant effect of climate change on winter haze pollution in Beijing, Atmos. Chem. Phys., 18, 17489-17496, 2018. Can the current work arbitrate based on the 20-year record? Probably not but it would be worth some comment.

Response: Thank you for your suggestion. We added the following discussion of these previous studies in line 274-282 "In the context of global warming, the unfavorable meteorological conditions in the northern part of China could be worsen in the future, although previous studies on the projection of the future effects of climate change on air pollution showed inconsistent results. For example, Cai et al. (2017) projected increased frequency and persistence of haze events in Beijing in the future (2050-2099) and Shen et al. (2018) found statistically insignificant trend of haze index in the future in Beijing. In contrast, in the southern part of China, especially in the YRD and surrounding regions, the estimated meteorological conditions were improving and were beneficial to pollution control (Chen et al., 2019). Further studies are needed to better understand the long-term trend of meteorological and climate effects on air pollution across China. Stricter clean air actions are preferred to avoid haze events in the future, considering the considerable meteorological effects on air pollution."

4.    Line 290: the same north-south contrast in the association of PM2.5 with RH was found by Zhai et al. (2019), previously cited but worth citing here, because they explained this contrast differently in terms of the origins of high-RH air masses and the links to aqueous chemical production and deposition.

**Response: We added the following discussion and the suggested citation in line 309-312 "Zhai et al. (2019) also discussed the north-south contrast in the PM$_{2.5}$-humidity associations and indicated that the positive effects of humidity on PM$_{2.5}$ in the north were partly attributed to the favorable role of aqueous-phase aerosol chemistry in secondary PM$_{2.5}$ formation and the negative PM$_{2.5}$-humidity associations in the south were partly attributed to the precipitation related wet deposition."**

5.    Line 325: I think 'interannual' should be 'long-term trends'

**Response: We changed this word as suggested.**

6.    Lines 361-362: I don't understand 'First, the satellite retrievals exhibited an increasing prediction error when hindcasting historical pollution levels.' …and the related discussion.

**Response: We added more explanation to clarify this discussion and adjusted the sentences in line 385-391 as follows "First, as reported by previous studies (Xiao et al., 2018;Xue et al., 2019), the satellite-based PM$_{2.5}$ prediction model suffered from increasing prediction error when hindcasting historical pollution levels a long time before the model training time period. One reason could be that some unobserved parameters, e.g. PM$_{2.5}$ composition, modify the associations between PM$_{2.5}$ and predictors, leading to model overfitting. The satellite-driven PM$_{2.5}$ prediction model used in this study is a state-of-the-art prediction model with improved prediction accuracy for high-pollution events, but its hindcast prediction quality could be further improved to better describe the historical PM$_{2.5}$ spatiotemporal distribution."**

This work used a combination of machine learning model, statistical model and chemical transport model to quantify the contribution to PM2.5 variation from meteorological impacts and emission changes during 2000–2018. It is indicated that although emissions dominated the long-term PM2.5 trends, the meteorology-driven anomalies also played a crucial role in PM2.5 trends. Overall, this manuscript is well structured and well written. I think this work well fits the scope of this journal and it is suggested to be published after addressing the following issues.

**Response: We thank the referee for the positive tone and the valuable suggestions to improve our manuscript.**

The authors emphasized the contribution of meteorology to interannual and seasonal trends of PM2.5, especially in fall and winter. Though some existing studies have conducted similar analysis, it would be more interesting to discuss the different meteorological factors in detail based on this GAM model, rather than summarizing as meteorological effects. Also, the mechanism of meteorological impact on PM2.5 might be quite different in the cold and warm season. It is also worth being analyzed since that the seasonal variation of PM2.5 is discussed here.

**Response: Thank you for these suggestions. We fitted regressions with normalized meteorology parameters and discussed their relative contributions in line 314-319, "Regarding the relative contribution of the different meteorology parameters, we found that over the south coast region, temperature and humidity showed greater effects than did the boundary layer height and precipitation. In winter, humidity, boundary layer height and precipitation were critical for the PM$_{2.5}$ variations in the middle and north of China. In summer and fall, the temperature and humidity were critical for the PM$_{2.5}$ variations across southern China. In spring, the temperature showed notable effects in the south coast region, and the precipitation exhibited large effects in the North China Plain." We summarized seasonal differences in meteorological effects on PM$_{2.5}$ in Figure A4 and in line 302-314, "Consistent with previous studies, we also observed spatially and seasonally varying associations between PM$_{2.5}$ and meteorological parameters that reflect the varying PM$_{2.5}$ responses to meteorological changes (Fig. A4). Temperature was positively associated with PM$_{2.5}$ in spring, summer and fall across East China; however, in winter, the temperature was negatively associated with PM$_{2.5}$ in northern China (He and Wang, 2017;Qiu et al., 2015) due to the low-temperature-related stable atmosphere and decreased evaporation loss of PM$_{2.5}$. Humidity yielded positive effects in northern China and negative effects in southern China in all seasons, especially in winter (He et al., 2017;Zhai et al., 2019). The spatial difference in the effects of humidity on PM$_{2.5}$ may occur due to a threshold of the humidity altering the direction of the humidity influence, from hygroscopic increase to wet deposition. Zhai et al. (2019) also discussed the north-south contrast in the PM$_{2.5}$-humidity associations and indicated that the positive effects of humidity on PM$_{2.5}$ in the north were partly attributed to the favorable role of aqueous-phase aerosol chemistry in secondary PM$_{2.5}$ formation and the negative PM$_{2.5}$-humidity associations in the south were partly attributed to the precipitation related wet deposition. The boundary height and precipitation were negatively associated with PM$_{2.5}$**

**across East China in all seasons, and the effect of precipitation was greater in northern China than that in southern China (Wang and Chen, 2016).” We also added discussion on the mechanisms of seasonal variations in meteorological impacts in line 319-321, “The seasonal variations in meteorological impacts could be due to the interactions between meteorological parameters that showed significant seasonal patterns. Further studies are needed to understand the mechanism of seasonal differences in the meteorology-pollution relationships.”**

Section 2.2 Although similar methods have been applied before, it is suggested to specify and justify the methodology and parameters used in this work. Some explanation is still needed. For example, why the PM2.5 concentration is correlated to wind at 500 hPa but some other work (e.g., Zhai et al., 2019) chose 850hPa.

**Response: We selected these meteorology parameters since they are previously reported significantly affect air pollution (Chen et al., 2018;Chen et al., 2020) and they contributed critically in previous PM$_{2.5}$ prediction models (She et al., 2020;Xiao et al., 2018). Specifically, we selected wind at 500 hPa rather than wind at 850 hPa since wind at 500 hPa is used to characterize air stagnation (Feng et al., 2020) and it performed significantly in the GAM model. It is notable that these meteorology parameters are correlated with each other (Cai et al., 2017) and it is hard to analyze the effects of individual meteorological factor with statistical methods. We added the following sentences to clarify the parameter selection in line 150-152, “These meteorological parameters have been reported to be strongly associated with PM$_{2.5}$ concentrations in various regions in China (Chen et al., 2020;Feng et al., 2020) and contributed significantly in previous PM$_{2.5}$ prediction models (She et al., 2020).”**

Line 143, missing “V wind at 500 hPa"?

**Response: We added these missing words.**

Section 3.1 and 3.2 are too short to be an individual section.

**Response: Thank you for this suggestion. We combined these two sections as Section 3.1.**

Line 165: delete the redundant reference "Maji et al., 2019 "

**Response: We deleted the repeated references.**

Line 209, change to "interannual variability" or "long-term trends”. It needs to be checked and corrected throughout the manuscript.

**Response: We changed the section title to “Interannual and seasonal variabilities of meteorology-associated PM$_{2.5}$”. We also reviewed the manuscript and corrected the related error.**

References:

[revised manuscript text omitted]